# Sliding Mode Control with Sliding Perturbation Observer-Based Strategy for Reducing Scratch Formation in Hot Rolling Process †

Hyun-Hee Kim [1], Sung-Jin Kim [1], Sung-Min Yoon [2], Yong-Joon Choi [3] and Min-Cheol Lee [4],*

1   Division of Robotics Convergence, Pusan National University, 2, Busandaehak-ro 63beon-gil, Geumjeong-gu, Busan 46241, Korea; sleepingjongmo@gmail.com (H.-H.K.); jins2410@gmail.com (S.-J.K.)
2   Ronfic Co., Inc., 34, Yutongdanji 1-ro 57beon-gil, Gangseo-gu, Busan 46721, Korea; tactics1019@gmail.com
3   POSCO Technology Research Lab, 6267, Donghaean-ro, Nam-gu, Pohang-si 37859, Korea; cyj@posco.com
4   School of Mechanical Engineering, Pusan National University, 2, Busandaehak-ro 63beon-gil, Geumjeong-gu, Busan 46241, Korea
*   Correspondence: mclee@pusan.ac.kr
†   Submission is extension of conference paper: Kim, H.-H.; Lee, M.-C. The Finite Element Analysis of Flying Touch Hot Rolling Method Using Ansys Simulation. In Proceedings of the 10th Asian Control Conference (ASCC), Sabah, Malaysia, 31 May–3 June 2015; doi:10.1109/ASCC.2015.7244667.

**Abstract:** In a hot rolling process, excessive friction between rollers and steel plates may lead to the formation of scratches on the steel plate. To reduce scratch formation in the finishing mill of the hot rolling process, two techniques are proposed in this work: flying touch and velocity synchronization. The proposed flying touch method can reduce the impact of the generated force when the upper roller collides with the steel plate. In addition, the proposed velocity synchronization method can decrease the frictional force resulting from the velocity difference between the rollers and steel plate. The effectiveness of the proposed methods was demonstrated through simulations and experiments using a 1/40 downscaled hot rolling simulator. The simulations and experimental results demonstrate that the proposed methods can reduce the magnitudes of friction and impact forces that lead to scratch formation on the steel plates in the hot rolling process.

**Keywords:** hot rolling; scratch; flying touch; velocity synchronization; downscaled simulator

## 1. Introduction

Manufacturers employ the hot rolling to process thick steel slabs into relatively thin steel coils, which consumers can conveniently use. Hence, reducing coil defects is crucial for improving the productivity and reliability of the steel industry. There are two types of coil defects: shape defect (caused by the rolling process) and quality defect (associated with the steel material) [1]. Shape defects include crowns, width failures, and thickness failures, as well as scratches caused by the intense frictional force between the rollers and steel plate.

Several studies have been conducted to reduce defects resulting from the rolling process. For example, Sun used the random forest method to predict the occurrence of strip crown defects during the hot rolling process [2]. Zhang discussed the macroscopic defects that occur during the superplastic formation and diffusion bonding of the four-layer structure of Ti–6Al–4V metal [3]. Kim examined the burr defects caused by the rolling process through a shear mechanism analysis and subsequently proposed a method to reduce such flaws through field experiments [4]. Research on friction, which is the main cause of defects in industrial automation, has also been conducted. Hwang compared various intelligent friction compensation methods based on motion control systems to reduce the frictional force that can cause defects in industrial products [5]. Kang implemented a frictional isolator in a rotary system to realize a high-precision roll-to-roll manufacturing process,

thus mitigating the undesirable effects of friction during continuous plastic film printing processes [6]. The study demonstrated that a friction-isolated rotary system can reduce the root-mean-square tracking error by up to 61% compared with the error when using a conventional roll-to-roll system without a frictional isolator.

In particular, studies on scratch defects caused by the intense friction between the rollers and steel plates during the rolling process have been conducted [7,8]. Chen and Li investigated the control methods for reducing scratch formation during the cold rolling process [7]. They reported that the generation of coil scratches can be reduced by adjusting parameters such as oil film thickness and temperature. Zhou studied the scratch resistance of steel during the hot rolling process based on the material properties of the roller [8]. The study confirmed that the microstructure of rollers can also affect scratch formation. The foregoing techniques can aid in diminishing the generation of scratches during hot rolling. However, to achieve better scratch formation reduction, it is necessary to develop designs involving mechanical and control methods that are capable of directly reducing the impact force.

In this study, the flying touch and velocity synchronization methods are proposed to achieve the following goals during the hot rolling process.

- Flying touch method

In the general hot rolling system, the steel plate passes between the fixed upper and lower rollers to process the plate into coils. In the proposed flying touch hot rolling method, the upper roller only lightly touches the steel plate upon contact, thus reducing the generated impact force. Accordingly, the roller design must enable the upper roller to move up and down.

- Velocity synchronization method

When the steel plate passes through the gap between the upper and lower rollers, plate scratching can occur because of the frictional force caused by the velocity difference between the rollers and steel plate. In the finishing mill, if the velocity of the steel plate is synchronized with the rotational velocity of the roller, then the frictional force during processing is diminished; consequently, reducing the probability of generating scratches.

The proposed velocity synchronization and flying touch methods for reducing the impact of friction between the rollers and steel plate during the hot rolling process are introduced in Section 2. To evaluate these two methods experimentally, the hot rolling system should be designed such that the upper rolling roller is moved vertically while the upper and lower rolling rollers are rotated. This proposed hot rolling system is built as a 1/40 downscaled model of an actual system operating in the Pohang Iron and Steel Company (POSCO, Pohang-si, Korea). The constructed hot rolling system is also utilized to evaluate the proposed scratch formation reduction algorithm formulated in this work.

To implement the velocity synchronization method, the feeding velocity of the steel plate must be measured; however, this is difficult to implement because of the environmental factors involved in the actual hot rolling process. In view of this, the technique for determining the frictional force between the roller and steel plate was indirectly applied to attain velocity synchronization. To estimate this force, a sliding perturbation observer (SPO) was implemented [9]. The SPO can estimate the reaction force and torque without attaching additional force/torque sensors. It is also combined with a sliding mode controller to robustly control the rotational velocity of the roller without chatter, even in severe environments such as that in the hot rolling process. Previous studies have applied the SPO to a surgical robot [10–12], hydraulic servo system [13–16], and robot manipulator [17,18] to estimate the reaction force without using a force sensor.

Section 2 elaborates on the strategy for reducing scratch formation using the flying touch and velocity synchronization methods. Section 3 presents the design of the sliding mode control (SMC) with an SPO (SMCSPO) for regulating the downscaled hot rolling simulator. Section 4 describes the simulations using Robotics Lab and ANSYS software to evaluate the proposed methods for reducing scratch formation. Section 5 describes

experiments conducted using the downscaled hot rolling simulator to evaluate the scratch formation reduction algorithm. Finally, Section 6 presents the conclusions.

## 2. Scratch Formation Reduction Strategy

### 2.1. Flying Touch Method

In the traditional hot rolling process, the gap between the upper and lower rollers is fixed [19] as the steel plate passes through the rollers. To roll the plate, the gap between the two rollers must be smaller than the thickness of the steel plate. Consequently, the steel plate may be damaged owing to the generated large frictional force. This force is called impact drop and is one of the main causes of coil defects.

The flying touch method controls the gap between the two rollers to enable the smooth rolling of the steel plate, thus reducing the impact drop. A simple concept of the flying touch method is shown in Figure 1. In the traditional hot rolling process shown in Figure 1a, the steel plate must pass through the gap between two fixed rollers. However, in the hot rolling system based on the flying touch method shown in Figure 1b, the gap between the rollers is set to be sufficiently large to allow the plate to pass between the two rollers before rolling begins. Subsequently, when hot rolling starts, the gap between the two rollers is controlled such that the rollers can smoothly press the steel plate. The flying touch method reduces the impact drop, which can cause scratch formation, by alleviating the impact force between the steel plate and rollers. However, to implement this method, a system for operating the upper roller in a direction perpendicular to the rolling direction must be incorporated.

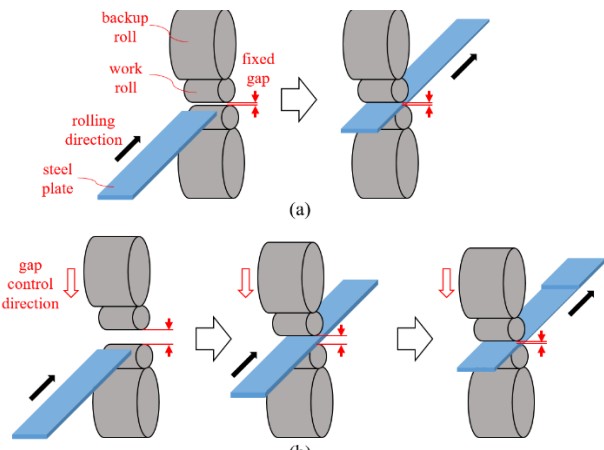

**Figure 1.** (**a**) Conventional hot rolling method in finishing mill; (**b**) hot rolling in finishing mill using flying touch method.

### 2.2. Velocity Synchronization

Figure 2 shows a system in which a motor is attached to the roller to synchronize the velocity of the rollers and steel plate. To reduce the frictional force generated when the steel plate and roller come into contact, the rotational velocity of the roller ($V_r$) and transfer velocity of the steel plate ($V_m$) must be synchronized, as shown in Figure 2. To achieve this, the velocities of the rollers and steel plate must be accurately measured. However, because of the harsh hot rolling process environment, attaching velocity sensors to the steel plate is challenging. Accordingly, instead of directly measuring the rotational and transfer velocities of the roller and steel plate, respectively, an indirect method of synchronizing these velocities is proposed.

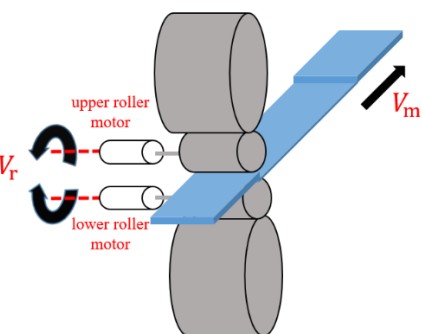

**Figure 2.** Hot rolling system configuration for velocity synchronization.

The frictional force exerted on the steel plate is caused by the velocity difference between the steel plate and rollers; this friction increases the load torque on the rotating motor of the roller. Therefore, if the load torque can be observed, then the friction force generated between the roller and steel plate can also be estimated. An SPO-based torque estimation method, which can estimate the load torque without using additional force/torque sensors, is used. To robustly control the system, the SMCSPO is applied to control the velocity of rollers.

Figure 3 illustrates the velocity synchronization concept between the rollers and steel plate. Throughout the manuscript, "ˆ" refers to estimated quantities. If the roller does not come in contact with the steel plate, its load torque ($\hat{T}_l$) is 0, and the rotational velocity ($V_r$) is equal to the roller's reference velocity ($V_{ref}$). However, if the roller is in contact with the steel plate, then $\hat{T}_l > 0$. This means that the rollers and steel plate velocities do not synchronize, possibly leading to the formation of scratches on the plate. In this case, the new reference velocity of the roller ($V_r$) is determined by the sum of the previous reference velocity of the roller ($V_{ref}$) and velocity compensation value ($V_c$), as follows:

$$V_r = V_{ref} + V_c, \tag{1}$$

where $V_c$ is proportional to $\hat{T}_l$ and determined by selecting the appropriate gain, $k_p$, as follows.

$$V_c = k_p \hat{T}_l. \tag{2}$$

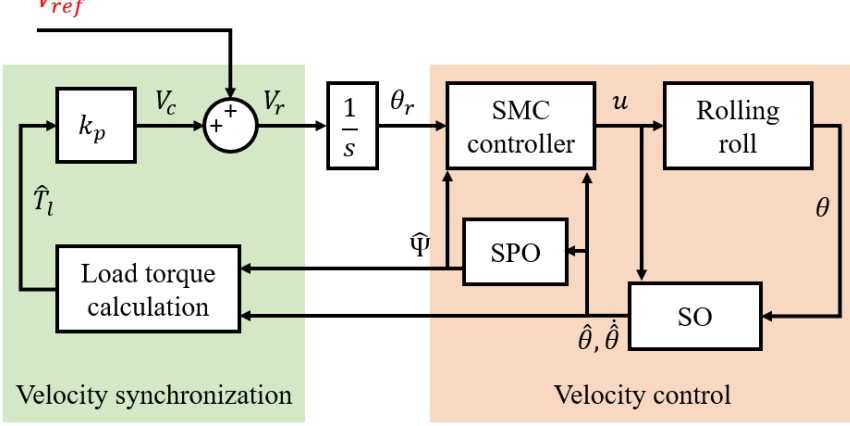

**Figure 3.** Velocity synchronization concept for steel plate and rollers using SMCSPO.

The velocity synchronization method for the rollers and steel plates can be implemented by selecting a suitable $k_p$ value.

### 3. Observer and Controller Design for Scratch Formation Reduction

*3.1. SPO*

In hot rolling systems, the gap control and rotation velocity control of the rollers should be capable of overcoming intense disturbances. The SMCSPO is a robust controller suitable for controlling nonlinear systems with disturbances. The SPO estimates the load torque of the roller's rotating motor for velocity synchronization; it also renders the SMCSPO controller to operate more robustly by estimating the nonlinear terms of the system to compensate for the control input. Furthermore, the perturbation estimated by the SPO can provide a standard for evaluating the probability of scratch formation.

In this section, the structure of an SPO is briefly discussed. The governing equation for the general second-order dynamics, including disturbance, is given as

$$\ddot{\theta} = f(\theta) + \Delta f(\theta) + (b(\theta) + \Delta b(\theta))u_c + d(t), \tag{3}$$

where $\theta$ is the rotation angle of the roller; $f(\theta)$ and $\Delta f(\theta)$ are the system dynamics matrices and their nonlinear terms, respectively; $b(\theta)$ and $\Delta b(\theta)$ are the control input matrices and their nonlinear terms, respectively; and $d(t)$ is the disturbance term caused by excessive friction between the roller and steel plate.

Perturbation is defined as the combination of all nonlinear terms and disturbances in (3), as follows:

$$\Psi = \Delta f(\theta) + \Delta b(\theta)u_c + d(t), \tag{4}$$

where the perturbations are assumed as upper-bounded by a known continuous function of state [20].

Based on the definition in (4), the nonlinear term is separated from the dynamics. For convenience, the linear term of the system is also separated. An arbitrary positive number ($\alpha_3$) and a new control variable ($\bar{u}$) are defined to decouple the control variable.

$$f(\hat{\theta}) + b(\hat{\theta})u_c = \alpha_3\bar{u}. \tag{5}$$

The state space representation is given by the following:

$$\begin{aligned} \dot{x}_1 &= x_2, \\ \dot{x}_2 &= \alpha_3\bar{u} + \Psi, \\ y &= x_1 = \theta. \end{aligned} \tag{6}$$

Using (4) and (5), (3) can be simplified as (6). A new state variable ($x_3$) for deriving the perturbation from the sliding observer is defined as

$$x_3 = \alpha_3 x_2 - \frac{\Psi}{\alpha_3}. \tag{7}$$

The perturbation is calculated using (7). The perturbation of the entire sliding observer structure [17] for monitoring the system states is given by

$$\begin{aligned} \dot{\hat{x}}_1 &= \hat{x}_2 - k_1 sat(\tilde{x}_1) - \alpha_1\tilde{x}_1, \\ \dot{\hat{x}}_2 &= \alpha_3\bar{u} - k_2 sat(\tilde{x}_1) - \alpha_2\tilde{x}_1 + \hat{\Psi}, \\ \dot{\hat{x}}_3 &= \alpha_3^2(-\hat{x}_3 + \alpha_3\hat{x}_2 + \bar{u}). \end{aligned} \tag{8}$$

The perturbation ($\hat{\Psi}$) can be calculated as

$$\hat{\Psi} = \alpha_3(-\hat{x}_3 + \alpha_3\hat{x}_2), \tag{9}$$

where $k_1$, $k_2$, $\alpha_1$, $\alpha_2$, and $\alpha_3$ are positive values, and $\widetilde{x}_i = \hat{x}_i - x_i$ is the estimated state error. Throughout the text, "~" refers to estimation errors. The saturation function is used to reduce chattering in the sliding surface and is given by

$$sat(\widetilde{x}_1) = \left\{ \begin{array}{l} \widetilde{x}_1/|\widetilde{x}_1|, \ if \ |\widetilde{x}_1| \geq \varepsilon_0 \\ \widetilde{x}_1/\varepsilon_0, \ if \ |\widetilde{x}_1| < \varepsilon_0 \end{array} \right. , \tag{10}$$

where $\varepsilon_0$ is the SPO boundary layer.

### 3.2. SMCSPO

This section presents the integration of the SMC and SPO schemes. For system (6), the estimated sliding function is defined as

$$\hat{s} = \dot{\hat{e}} + c\hat{e}, \tag{11}$$

where $c > 0$ and $\hat{e} = \hat{x}_1 - x_{1d}$ are the estimated position tracking errors. The control input, $\overline{u}$, is selected to enforce a sufficient condition for the existence of a sliding mode, $\hat{s}\dot{\hat{s}} < 0$. The desired $s$-dynamics is selected as

$$\dot{\hat{s}} = -K \, sat(\hat{s}), \tag{12}$$

where

$$sat(\hat{s}) = \left\{ \begin{array}{l} \hat{s}/|\hat{s}|, \ if \ |\hat{s}| \geq \varepsilon_c \\ \hat{s}/\varepsilon_c, \ if \ |\hat{s}| < \varepsilon_c \end{array} \right. . \tag{13}$$

The foregoing is utilized because of its desirable anti-chatter properties. In the equation, $\varepsilon_c$, similar to $\varepsilon_o$ in the SPO, represents the boundary layer of the SMC controller. Using the results of the previous sections, $\dot{\hat{s}}$ can be computed as

$$\dot{\hat{s}} = \alpha_3\overline{u} - \left[ \frac{k_2}{\varepsilon_o} + c\left(\frac{k_1}{\varepsilon_o}\right) - \left(\frac{k_1}{\varepsilon_o}\right)^2 \right]\widetilde{x}_1 - \left(\frac{k_1}{\varepsilon_o}\right)\widetilde{x}_2 - \ddot{x}_{1d} + c(\hat{x}_2 - \dot{x}_{1d}) + \hat{\Psi}. \tag{14}$$

When $\widetilde{x}_2 = 0$, a control law is selected using (12) and (14), as follows:

$$\overline{u} = \frac{1}{\alpha_3}\left\{ -K \, sat(\hat{s}) + \left[ \frac{k_2}{\varepsilon_o} + c\left(\frac{k_1}{\varepsilon_o}\right) - \left(\frac{k_1}{\varepsilon_o}\right)^2 \right]\widetilde{x}_1 + \ddot{x}_{1d} - c(\hat{x}_2 - \dot{x}_{1d}) - \hat{\Psi}. \tag{15}$$

### 3.3. Stability Analysis of SPO and SMCSPO

The stability of the SPO and SMCSPO were analyzed. As shown in (4), the perturbation consists of all the nonlinear terms of system dynamics. Each perturbation term is assumed to be upper-bounded by the known continuous functions of the state as follows [9]:

$$\Gamma(\theta, t) = F(\theta) + \Phi(\theta)u_c + D(t) > |\Psi(t)|, \tag{16}$$

where $F > |\Delta f|$, $\Phi > |\Delta b|$, and $D > |d|$ are the expected upper bounds of uncertain nonlinear terms. In the sliding observer, the conditions for the existence of the sliding mode can be expressed as

$$\begin{array}{l} \widetilde{\theta}_2 \leq k_1 + \alpha_1\widetilde{\theta}_1, \ (\text{if } \widetilde{\theta}_1 > 0), \\ \widetilde{\theta}_2 \geq -k_1 + \alpha_1\widetilde{\theta}_1, \ (\text{if } \widetilde{\theta}_1 < 0), \end{array} \tag{17}$$

where $\widetilde{\theta}_1 = \widetilde{x}_1$ and $\widetilde{\theta}_2 = \widetilde{x}_2$ are the estimation errors of the system representing the position and velocity, respectively. When the sliding mode occurs, the resulting error dynamics obtained by (6) and (8) assume the following equation:

$$\dot{\tilde{\theta}}_2 = -\Psi - \left(\frac{k_2}{k_1}\right)\tilde{\theta}_2. \tag{18}$$

The stability of the sliding observer, $\tilde{\theta}_2\dot{\tilde{\theta}}_2 < 0$, can be ensured by setting the following conditions: $k_2 > \Gamma$ and $\left|\tilde{\theta}_2\right| \leq k_1$. If $\tilde{\theta}_2 < 0$, then the stability is guaranteed and established by (19).

$$\dot{\tilde{\theta}}_2 = -\Psi - \left(\frac{k_2}{k_1}\right)\tilde{\theta}_2 \geq -\Psi + k_2 > 0 \tag{19}$$

Using a similar method, if $\tilde{\theta}_2 > 0$, then the stability is established by (20).

$$\dot{\tilde{\theta}}_2 = -\Psi - \left(\frac{k_2}{k_1}\right)\tilde{\theta}_2 \leq -\Psi - k_2 < 0, \tag{20}$$

where (19) and (20) ensure that $\tilde{\theta}_2\dot{\tilde{\theta}}_2 < 0$ by setting $k_2 > \Gamma$ and $\left|\tilde{\theta}_2\right| \leq k_1$. Therefore, if the gains ($k_1$ and $k_2$) are sufficiently large, the stability of the SPO can be guaranteed.

To check the closed-loop stability of the SMCSPO, the Lyapunov stability criterion is used, as follows:

$$\begin{aligned} V &= \tfrac{1}{2}\hat{s}^2 > 0, \\ \dot{V} &= \hat{s}\dot{\hat{s}} < 0. \end{aligned} \tag{21}$$

The estimated sliding surface of the SMCSPO is given by (11), which can also be expressed as

$$\hat{s} = \dot{\hat{\theta}}_1 - \dot{\theta}_{1d} + c\left(\hat{\theta}_1 - \theta_{1d}\right), \tag{22}$$

where $\theta_{1d}$ is the desired position, and $\dot{\hat{\theta}}_1 = \dot{\hat{x}}_1$. After the reaching phase, $\alpha_1$ and $\alpha_2$ are set as zero; therefore, (23) can be derived by combining (8) and (22).

$$\hat{s} = \hat{\theta}_2 - \left(\frac{k_1}{\varepsilon_0}\right)\tilde{\theta}_1 - \dot{\theta}_{1d} + c\left(\hat{\theta}_1 - \theta_{1d}\right) \tag{23}$$

Differentiating (23) yields (24), as follows:

$$\dot{\hat{s}} = \dot{\hat{\theta}}_2 - \left(\frac{k_1}{\varepsilon_0}\right)\dot{\tilde{\theta}}_1 - \ddot{\theta}_{1d} + c\left(\dot{\hat{\theta}}_1 - \dot{\theta}_{1d}\right). \tag{24}$$

Using the results presented in Section 3.1 and substituting (8) into (24), we obtain

$$\dot{\hat{s}} = \alpha_3\bar{u} - \frac{k_2}{\varepsilon_o}\tilde{\theta}_1 + \hat{\Psi} - \frac{k_1}{\varepsilon_o}\left[\tilde{\theta}_2 - \frac{k_1}{\varepsilon_o}\tilde{\theta}_1\right] - \ddot{\theta}_{1d} + c\left[\hat{\theta}_2 - \frac{k_1}{\varepsilon_o}\tilde{\theta}_1 - \dot{\theta}_{1d}\right], \tag{25}$$

where the new control variable ($\bar{u}$) is selected to enforce $\hat{s}\dot{\hat{s}} < 0$ outside a predefined manifold, $|\hat{s}| \leq \varepsilon_c$. Therefore, the substitution of (12) into (25) yields the following:

$$\bar{u} = \frac{1}{\alpha_3}\left\{-K\,sat(\hat{s}) + \left[\frac{k_2}{\varepsilon_o} + c\left(\frac{k_1}{\varepsilon_o}\right) - \left(\frac{k_1}{\varepsilon_o}\right)^2\right]\tilde{\theta}_1 + \ddot{\theta}_{1d} - c\left(\hat{\theta}_2 - \dot{\theta}_{1d}\right) - \hat{\Psi}. \tag{26}$$

If the effects of $\tilde{\theta}_2$ are considered, then (12) becomes

$$\dot{\hat{s}} = -K\,sat(\hat{s}) - \left(\frac{k_1}{\varepsilon_o}\right)\tilde{\theta}_2. \tag{27}$$

From the sliding condition in (17), the state estimation error is bounded by $\left|\tilde{\theta}_2\right| \leq k_1$.

To satisfy the Lyapunov stability criterion, $\dot{\hat{s}}\hat{s} < 0$, outside a predefined manifold, $|\hat{s}| \leq \varepsilon_c$, the SMCSPO control gain must be chosen such that

$$K > \frac{k_1^2}{\varepsilon_o} > \left| \left( \frac{k_1}{\varepsilon_o} \right) \widetilde{\theta}_2 \right|. \tag{28}$$

Therefore, if the gain ($K$) is sufficiently large, then the stability of the SMCSPO can be ensured.

## 4. Simulation

### 4.1. Robotics Lab Simulation

Robotics Lab, a simulation program, was used to evaluate the estimation performance of the SPO in hot rolling systems. It has its own dynamics and control engine [21] and consists of a user modeling tool (rBuilder), control simulator, and data visualization tool (rPlot). The user can directly develop the control and kinematic algorithms in Microsoft Visual C++ and then interlock these with Robotics Lab.

The simplified hot rolling model was designed using Robotics Lab, which was used to simulate the occurrence of the roller's perturbation when the rollers come into contact with the steel plate (Figure 4). The movement of the steel plate (green metal) between the two rollers is from left to right. The rotational and transfer velocities of the rollers and steel plate are denoted as $V_r$ and $V_m$, respectively. In this simulation, $V_r$ was set to $0.9V_m$ to produce a strong disturbance, and the perturbation caused by the force resulting from the collision of the roller and steel plate was estimated by the SPO.

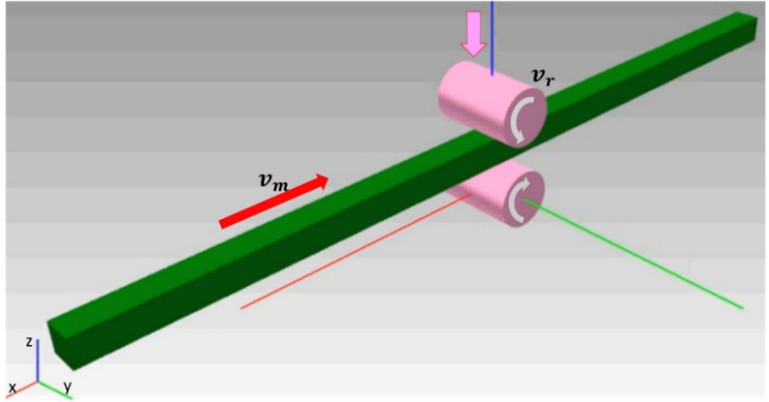

**Figure 4.** Robotics Lab modeling of simplified hot rolling system.

Figure 5 shows the estimated perturbation data using the SPO based on the encoder data of the motor rotating the upper roller. The roller collided with the steel plate at approximately 0.65 s after the initiation of simulation. Since the collision, a large perturbation value was consistently observed. The simulation result confirms that the assumed perturbation estimated by the SPO is the collision force disturbance between the steel plate and roller. The perturbation estimated by the SPO is used in practical experiments to minimize the friction between the roller and steel plate.

Figure 6 shows the moving velocity profile of the steel plate. The graph shows that the velocity fluctuates owing to the excessive frictional force generated after the roller collides with the steel plate. Therefore, the rotational velocity of the roller should be synchronized with the moving velocity of the steel plate using a velocity synchronization algorithm.

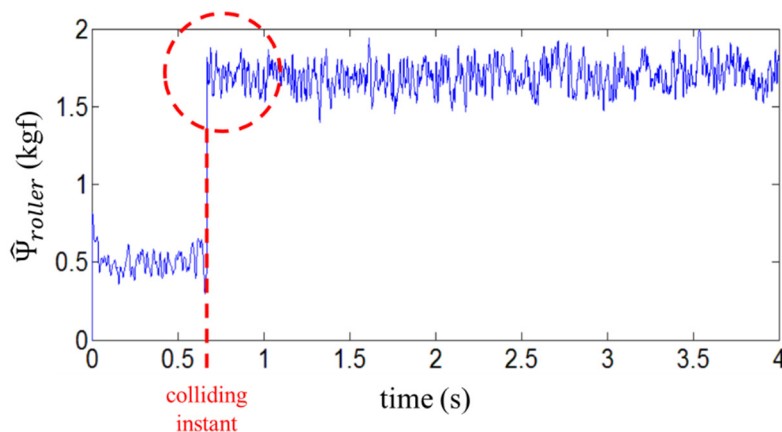

**Figure 5.** Graph of estimated perturbation of upper roller.

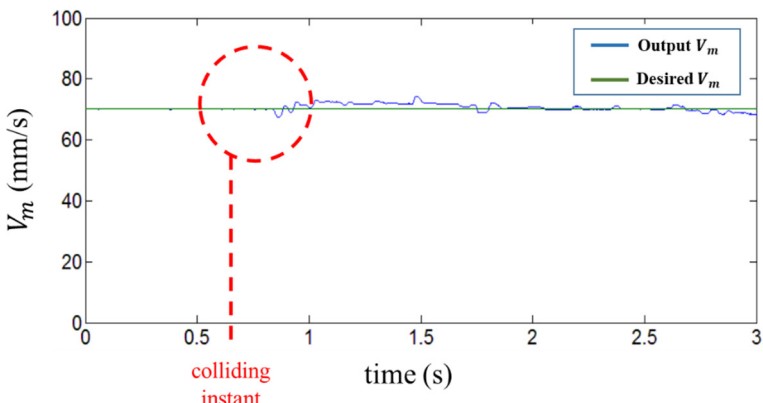

**Figure 6.** Output and desired velocities of steel plate during rolling.

## 4.2. ANSYS Simulation for Velocity Synchronization

The stress distribution during steel plate rolling can be simulated using a finite element analysis program. In this section, the stress distribution of the steel plate according to the descending profile of the roller is elaborated using the ANSYS workbench. A 1/4 size model of POSCO's actual hot rolling system, composed only of the upper roller and a steel plate with half of the usual thickness, is employed for stress analysis. The upper roller and steel plate modeled in the ANSYS workbench are shown in Figure 7.

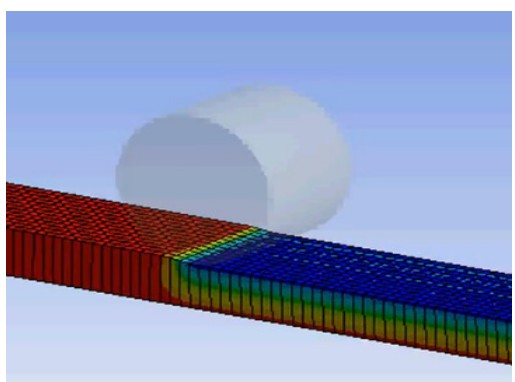

**Figure 7.** 1/4 size hot rolling simulator.

The materials of the roller and steel plate modeled in the simulation were designed based on the system used in actual hot rolling processes (POSCO, Pohang-si, Korea). The parameters of the stress analysis simulations are listed in Table 1. This simulation indicates that the friction coefficient increases if the roller velocities do not synchronize with that

of the steel plate. This is because the finite element analysis program assumes that the velocity of the roller is equal to that of the steel plate at the moment of contact.

**Table 1.** Properties of stress analysis simulation.

| Properties | Values |
|---|---|
| Thickness of the steel plate (half size) | 200 mm |
| Diameter of the upper roller | 630 mm |
| Transfer velocity of the steel plate | 500 mm/s |
| Rolled thickness of the steel plate | 12 mm |
| Initial gap between the upper roller and the steel plate | 20 mm |
| Simulation duration | 10 s |

Figure 8 shows the change in the steel plate stress over time when the friction coefficients are 0.2 and 0.1. This stress is generated by the friction force resulting from the velocity difference between the upper roller and steel plate. The upper roller starts to roll the plate at the beginning of the simulation. When the friction coefficients are 0.2 and 0.1, the maximum stresses are 370.13 and 326.29 MPa, respectively; furthermore, the stress was more evenly distributed when the friction coefficient was 0.1. Consequently, the stress is greater when the roller velocity does not synchronize well with that of the steel plate, causing the formation of more scratches during the rolling process.

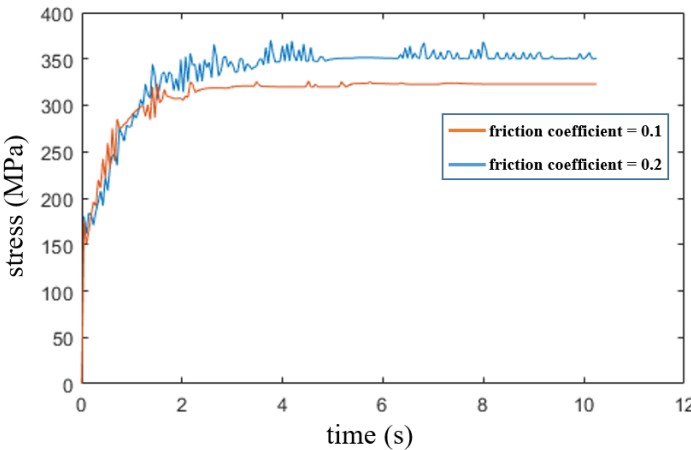

**Figure 8.** Distribution of stress according to velocity difference between roller and steel plate.

### 4.3. ANSYS Simulation for Flying Touch

The gap between the two rollers in the flying touch simulation can only be controlled by the upper roller (the lower roller is fixed); hence, the simulation in this research is implemented according to the velocity profiles of the upper roller only.

Two types of tests were simultaneously performed in the roller gap control simulation. The first test was conducted to determine the effect of the descending velocity of the roller on the impact force exerted on the steel plate. The second test compared the cases in which the roller descended at a constant speed with the case in which the roller gradually decelerated. In these simulations, the roller velocity was assumed to be well-synchronized with the steel plate velocity to evaluate the effects of the descending velocity of the roller only.

Figure 9 shows the simulation results of the change in stress when the descending velocities of the roller are constant at 20 and 40 mm/s. The simulation results indicate that, immediately after the initiation of rolling, the stress caused by the descending velocity of 40 mm/s was seemingly greater than that due to the stress caused by a descending velocity of 20 mm/s. Although the difference between the two descending speeds did not have a significant effect on the stress during the continuous rolling period after the roller collided with the steel plate, the downward motion of the roller with a low descending velocity, to the extent feasible to reduce stress at the instant of collision, was confirmed to be better.

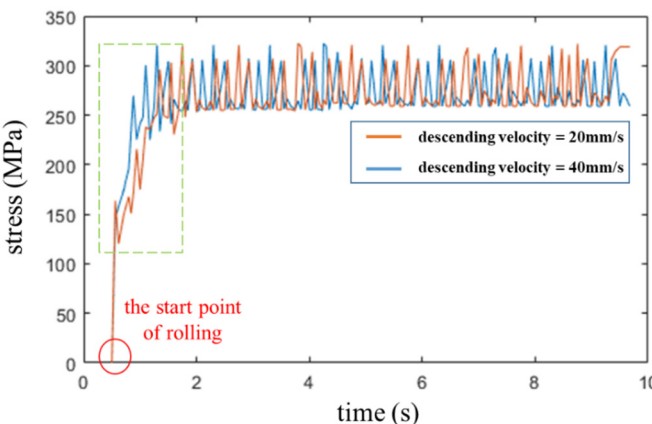

**Figure 9.** Stress distribution when roller descends at constant velocities (20 and 40 mm/s).

In the second test, the roller is given a trajectory that followed a curved velocity profile to descend smoothly. For comparison with the first test, the instantaneous velocities when the roller collides with the steel plate are assumed to be 20 and 40 mm/s. The experimental results are shown in Figure 10, where the area marked with a red circle in the graph indicates the moment of impact. The stress graph shows that the stress increases starting from the time of impact.

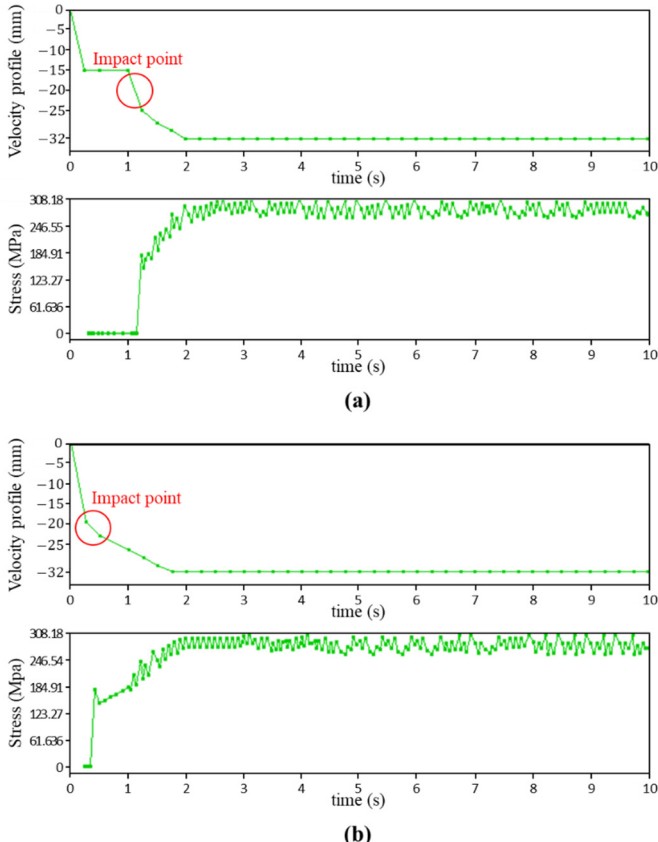

**Figure 10.** Stress distribution when roller descends according to curved velocity profile. Graphs of velocity profile of upper roller and steel plate stress when descending velocities are (**a**) 40 mm/s and (**b**) 20 mm/s at moment of impact.

The results of the experiment shown in Figures 9 and 10 are summarized in Table 2. The stress when the descending roller was gradually decelerated until the moment of collision appeared to be less than that when the roller had a constant velocity. Furthermore,

it was confirmed that the change in the amount of stress at the moment of impact when applying the proposed flying touch velocity profile (Figure 10) was considerably smaller than that when a constant velocity was applied. This phenomenon intensifies in the case of actual hot rolling because of the disturbance and vibration present in the real environment. The positional trajectory of the third-order equation was created based on the decelerated descending profile (20 mm/s) to apply the curved descending velocity trajectory to the vertical motion of motor of the upper roller.

**Table 2.** Stress simulation results.

| Properties | Constant Velocity of Descending Roller (MPa) | | Decelerating Velocity of Descending Roller (Mpa) | |
|---|---|---|---|---|
| | 40 mm/s | 20 mm/s | 40 mm/s | 20 mm/s |
| Total average stress on the steel plate | 290.24 | 289.47 | 284.80 | 284.29 |
| Maximum stress on the steel plate | 322.53 | 322.53 | 308.13 | 308.13 |
| Standard deviation stress after impact point | 23.29 | 24.40 | 12.46 | 13.54 |
| Stress on the steel plate at the moment of impact | 269.49 | 163.53 | 181.68 | 181.16 |

## 5. Experiment

### 5.1. Experimental System Configuration

The flying touch method cannot be applied to a general real hot rolling simulator because the vertical motion of the upper roller cannot be realized in a real system. Therefore, a downscaled simulator was designed for such an application. Different from the existing rolling system in which the upper roller is fixed, the upper roller of the designed simulator is capable of moving in the vertical direction. As shown in Figure 11, a single-axis actuator is designed to enable the upper roller to move vertically using a motor and a ball screw [22].

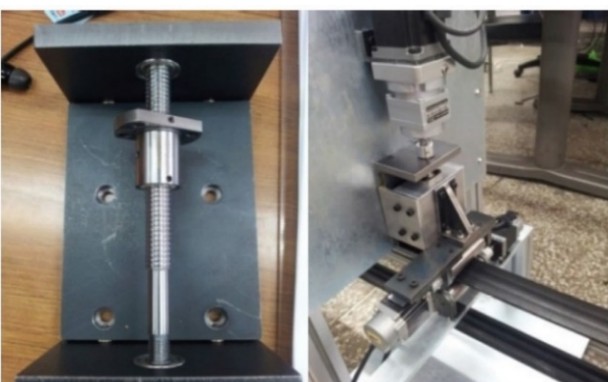

**Figure 11.** Mechanism for vertical movement of upper roller.

The designed simulator, which uses an electric motor, is characterized by its inability to generate sufficient torque; consequently, rolling the actual steel plate is difficult. To mitigate this, the experiment was performed using a rubber belt instead of a metal slab. The overall structure of the 1/40 downscaled hot rolling simulator for implementing the scratch formation reduction algorithm is shown in Figure 12.

A simulator with four motors was used for the rudimentary implementation of the rolling process. Motor M1, which allows the upper roller to move up and down in a vertical direction, is an actuator that implements the flying touch method. Motor M2 continuously rotates clockwise during rolling to transport the rubber belt, allowing it to pass between the upper and lower rollers. Motors M3 and M4 rotate the upper and lower rollers, respectively. These motors were used to implement the velocity synchronization algorithm. Each motor was appropriately selected by considering the expected load during rolling. In addition, all

motors in this simulator were controlled by the robust control algorithm of the SMCSPO to overcome the disturbances and nonlinearities in the rolling process. The continuous torque and rated power of the motors, the size of the simulator, and the materials are listed in Table 3. Although the simulator shape reflects that of the POSCO hot rolling system, its size is 1/40 times that of the real system.

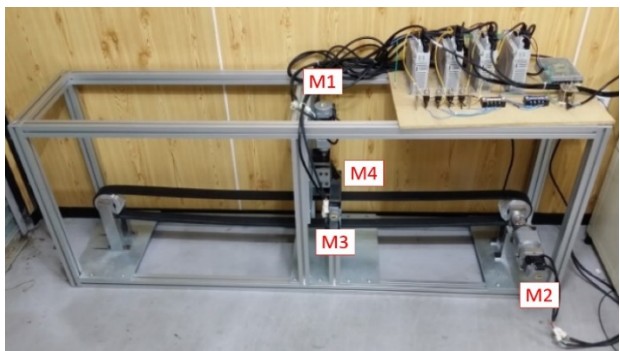

**Figure 12.** Hot rolling simulator for scratch formation reduction experiment.

**Table 3.** Specifications of experimental system.

| Properties | | Values |
|---|---|---|
| Continuous torque of motors (rated power) | M1 | 0.640 Nm (200 W) |
| | M2 | 0.640 Nm (200 W) |
| | M3, M4 | 0.318 Nm (100 W) |
| Size of the simulator | Length (rolling direction) | 1780 mm |
| | Width | 410 mm |
| | Height | 700 mm |
| Material of the simulator | | Steel and aluminum |

The detailed control system configuration for the downscaled hot rolling simulator is shown in Figure 13. Each motor is connected to an alternating current (AC) servo driver that provides encoder data and control input to the motor. The personal computer affords control commands to the servo driver using an multi motion controller (MMC) board, which converts digital signals to analog signals.

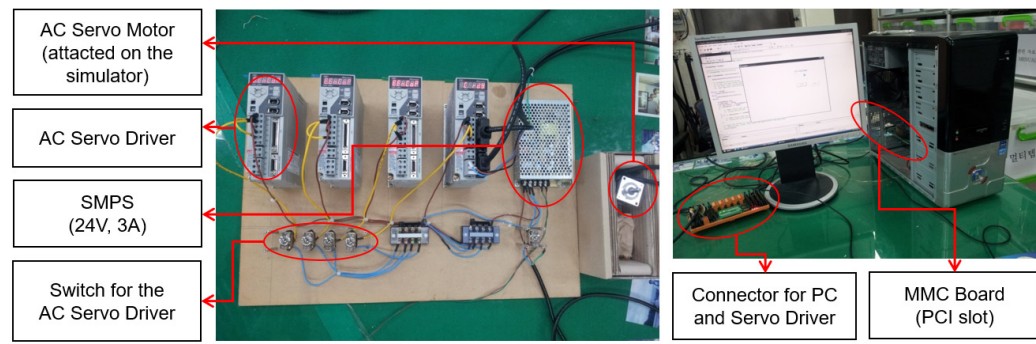

**Figure 13.** Control System configuration for downscaled hot rolling simulator.

*5.2. Scratch Formation Reduction Algorithm Experiment*

The experiment was performed sequentially using the following steps. First, the upper roller was raised 14 mm above the rubber belt. Second, motor M2 for transporting the rubber belt was operated clockwise. Finally, the upper roller was lowered by 15 mm to press the rubber belt by 1 mm.

Figure 14 compares the experimental results with and without the curved velocity profile of M1 and the application of the velocity synchronization method. The vertical

blue and red dotted lines in the graph denote the instances when the upper roller collides with the rubber belt in cases when the proposed algorithm is applied and not applied, respectively. In the case of the experiment in which the scratch formation reduction algorithm was not implemented, the velocity immediately became constant because the decelerated velocity profile was not applied. Therefore, the colliding instant was relatively earlier than the instant with the application of the scratch formation reduction algorithm.

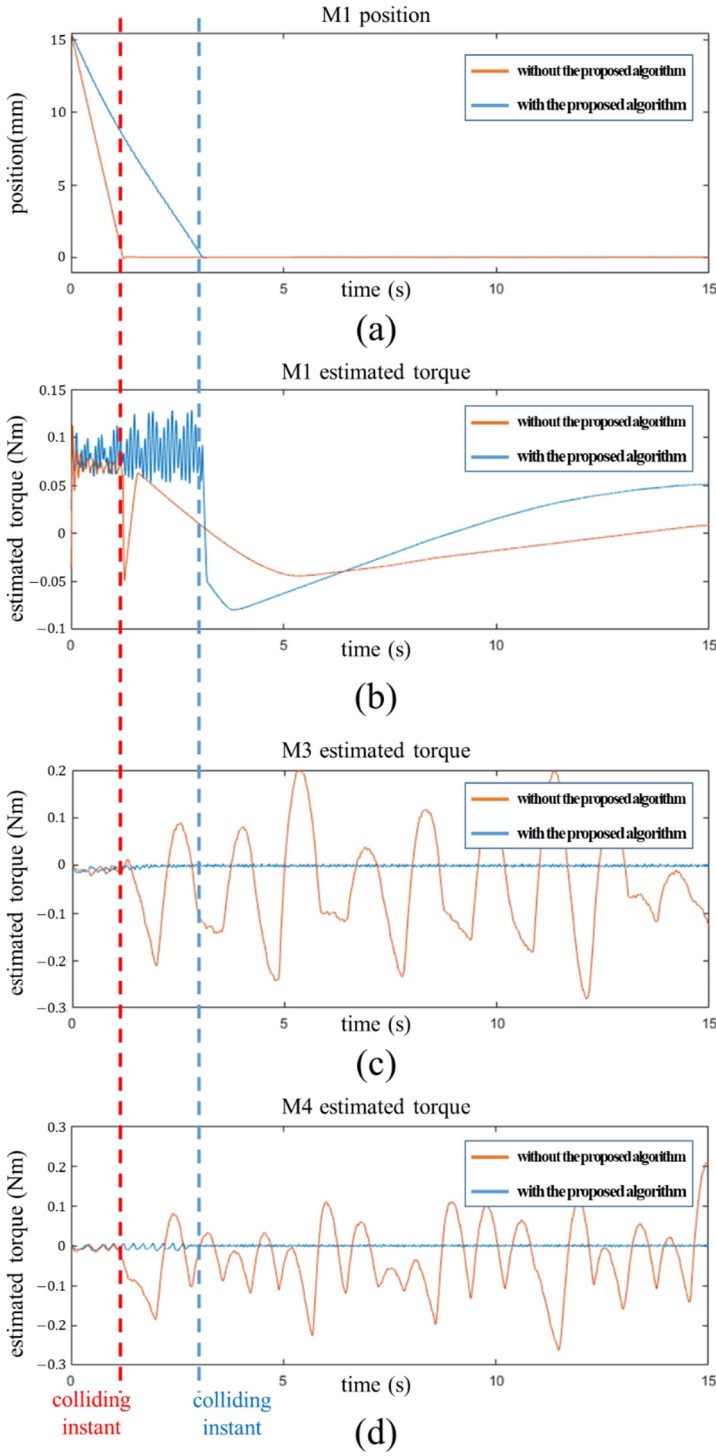

**Figure 14.** Comparison of experimental results when flying touch and velocity synchronization methods are applied or not applied. (**a**) M1 position; (**b**) M1 estimated torque; (**c**) M3 estimated torque; and (**d**) M4 estimated torque.

Figure 14b shows the estimated reaction torque when the M1 motor descends to the trajectory shown in Figure 14a. From the instant after collision (red line in Figure 14b), the estimated torque significantly fluctuated because the scratch reduction algorithm was not applied. When the scratch reduction algorithm was applied, the change in the estimated reaction force was relatively smooth at the moment of collision. The estimated reaction torque results of motors M3 and M4 are shown in Figure 14c,d, respectively, and summarized in Table 4. In the experiment in which the proposed algorithm was applied, the maximum torque, mean torque, and standard deviation values was considerably smaller than the experiment where the algorithm was not applied. If the reaction torque estimated by the SPO is small, the roller rotational velocity and steel plate velocity are considered well-synchronized. This means that the frictional force generated between the steel plate and rollers is reduced.

**Table 4.** Estimated torques of M3 and M4.

| Properties | With the Proposed Algorithm (Nm) | | Without the Proposed Algorithm (Nm) | |
|---|---|---|---|---|
| | M3 | M4 | M3 | M4 |
| Maximum torque | 0.0057 | 0.0040 | 0.2079 | 0.1997 |
| Mean torque | $-0.0012$ | $-0.0012$ | $-0.0306$ | $-0.0360$ |
| Standard deviation | 0.0030 | 0.0030 | 0.0795 | 0.1045 |

## 6. Conclusions

In this paper, two techniques were proposed to reduce the formation of scratches caused by hot rolling: flying touch and velocity synchronization. Through simulations and experiments, the use of the flying touch method based on a curved velocity profile was confirmed to smooth the change in the magnitude of the impact force at the moment of collision, as compared with the case of application of constant velocity. Moreover, the velocity synchronization algorithm was found capable of reducing the frictional force between the roller and steel plate by reducing the estimated reaction torque in the SPO. Reducing the friction and impact force through the two proposed methods can contribute to the decrease in the probability of scratch formation in the steel plate during the hot rolling process.

However, owing to the limitations of the downscaled simulator, confirming whether the proposed methods can reduce the formation of scratches on an actual steel plate in a real hot rolling system is exigent. In the future, further studies should be conducted to apply these proposed methods to a real system.

**Author Contributions:** Conceptualization, H.-H.K., Y.-J.C. and M.-C.L.; data curation, H.-H.K.; methodology, S.-J.K., S.-M.Y. and Y.-J.C.; project administration, M.-C.L.; software, S.-M.Y.; supervision, Y.-J.C. and M.-C.L.; validation, H.-H.K., S.-J.K. and S.-M.Y.; writing—original draft, H.-H.K.; writing—review & editing, M.-C.L. All authors have read and agreed to the published version of the manuscript.

**Funding:** This research was supported by the Technology Innovation Program (10073147, Development of Robot Manipulation Technology by Using Artificial Intelligence) funded By the Ministry of Trade, Industry & Energy (MOTIE, Korea), and supported in part by the nuclear research and development program through the National Research Foundation of Korea (NRF), funded by the Ministry of Science and ICT (MSIT, Korea) under Grant NRF-2019M2C9A1057807.

**Institutional Review Board Statement:** Not applicable.

**Informed Consent Statement:** Not applicable.

**Data Availability Statement:** The data presented in this study are openly available in follow link. https://drive.google.com/file/d/1mChSkaeVXHUINT9axcfcYNG_M5U5HgVK/view?usp=sharing (accessed on 14 April 2021).

**Conflicts of Interest:** The authors declare no conflict of interest.

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
