# Peer review of "Sliding Mode Control with Sliding Perturbation Observer-Based Strategy for Reducing Scratch Formation in Hot Rolling Process†"

_applsci, doi:10.3390/app11125526_

Round 1

Reviewer 1 Report

The authors presents a sort of control for "compensating" the effect of frictions.

Overall, the paper does not present any novelty but it has the merit to propose an interesting study. It is commendable that the authors give simulations and experimental results.

Perhaps they could improve their analysis because I found the a bit too descriptive.

Reviewer 2 Report

In this paper, the authors propose two methods for solving  scratch issue. This problem is quite important because it may result in the damaged products. The scratches are caused by excessive friction between the rolling roller and the steel plate. The authors present two ways to minimize the scratches: one is to design a flying touch method, and the other one is to use a velocity controller. The first method reduces the impact force generated between the roller and steel plate, while the second method can minimize the friction effects.  The authors use the simulation results to illustrated the effectiveness of the proposed methods.

Theoretically speaking, the flying touch method is not new. The authors consider both flying touch and velocity control for solving the scratches and this may be the main point of the contribution.   However, theoretical analysis is not enough.

Technically speaking, the authors give the detailed design process. It seems that readers can follow their methods for solving the scratches. However, the practical consideration is not given.

Please consider the following points when revising the paper.

  1. The authors may cite the following papers when discussing frictional issues.

     1)Intelligent friction compensation: A review. S Huang, W Liang, KK Tan

IEEE/ASME Transactions on Mechatronics 24 (4), 1763-1774     2) Friction isolated rotary system for high-precision roll-to-roll manufacturing, D.Kang, X. Dong,H.Kim, P.Park, C.E.Okwudire,Precision Engineering,Volume 68, March 2021, Pages 358-364
      2. Flying touch method: the authors should give some details regarding the design. It is better to give an example for the design, including motor and material sizes etc.
  1. Velocity control method: Is Eq(3) observer?   If so, I don't find the feedback information to correct your observer. Please check it. Some theoretical analysis may be discussed if possible. For example, the convergence analysis.
  2. Still velocity control issues. The control (15) is your controller. It is a sliding mode control. Could you give an analysis, for example, the stability analysis of the closed-loop system?  If you do so, this can give readers the confidence using your controller.
  3.  In your experimental tests, could you give some details regarding the configuration? You may draw several figures or blocks to illustrate your systems.

Round 2

Reviewer 2 Report

In this paper, the authors use  flying touch method and the velocity synchronization method  to minimize scratches during the hot rolling process.

This version is quite good. The authors have answered me all questions following my comments.  Specifically, the authors give a rigid proof regarding the closed-loop stability. I appreciate the authors' efforts.